# Dissolving salt is not equivalent to applying a pressure on water

Chunyi Zhang [1], Shuwen Yue [2], Athanassios Z. Panagiotopoulos[2], Michael L. Klein [1,3,4✉] & Xifan Wu [1✉]

Salt water is ubiquitous, playing crucial roles in geological and physiological processes. Despite centuries of investigations, whether or not water's structure is drastically changed by dissolved ions is still debated. Based on density functional theory, we employ machine learning based molecular dynamics to model sodium chloride, potassium chloride, and sodium bromide solutions at different concentrations. The resulting reciprocal-space structure factors agree quantitatively with neutron diffraction data. Here we provide clear evidence that the ions in salt water do not distort the structure of water in the same way as neat water responds to elevated pressure. Rather, the computed structural changes are restricted to the ionic first solvation shells intruding into the hydrogen bond network, beyond which the oxygen radial-distribution function does not undergo major change relative to neat water. Our findings suggest that the widely cited pressure-like effect on the solvent in Hofmeister series ionic solutions should be carefully revisited.

[1] Department of Physics, Temple University, Philadelphia, PA 19122, USA. [2] Department of Chemical and Biological Engineering, Princeton University, Princeton, NJ 08544, USA. [3] Institute for Computational Molecular Science, Temple University, Philadelphia, PA 19122, USA. [4] Department of Chemistry, Temple University, Philadelphia, PA 19122, USA. ✉email: mlklein@temple.edu; xifanwu@temple.edu

For over a century, scientists have been puzzled by the precise nature of electrolyte solutions[1–14]. Experimental studies, conducted in the late 19th century by Hofmeister, revealed that ions have diverse efficiencies in salting-out proteins from egg whites and blood serum[15]. Gurney explained Hofmeister's observations by the variable ability among ions to modify the structure of water: ions that strengthen the hydrogen-bond network of water were categorized as structure "makers", whereas other ions, called structure "breakers", weaken the structure[16]. Thereafter, the structure maker/breaker theory became textbook material and was widely applied to explain phenomena in electrolyte solutions[10]. In recent decades, insightful neutron diffraction experimental measurements became available for electrolytes[2–4]. The resulting structural information, as a function of concentration, exhibited a striking resemblance to the behavior of neat water under increasing pressure[4,17,18]. Seemingly, the hydrogen-bond network of water in sodium chloride (NaCl) and potassium chloride (KCl) solutions is distorted in a fashion equivalent to pure water under a few thousand atmospheres pressure[4,17,18]. This interpretation of neutron diffraction experiments challenges the structure maker/breaker theory because NaCl has a relatively small effect among the Hofmeister ions[19]. More recently, the ultrafast infrared pump-probe experiment revealed a completely different picture[7]. Namely, the dynamics of water molecules outside ionic first solvation shells (FSSs) retain their bulk properties as in the case of neat water[7]. The localized effect on the water structure by dissolved monovalent ions was further substantiated by dielectric relaxation spectroscopy experiments[11].

The contradictory pictures of electrolyte structure have been debated extensively as more experimental and theoretical work contributed new insights[8,20–29]. Compared to the dynamical information measured by vibrational spectroscopy[7], the liquid structure can be directly probed by diffraction experiments[4,17,18]. In neutron diffraction from electrolyte solutions, necessary empirical fitting[17,18] brings uncertainty into the interpretation of the data[30]. Moreover, theoretical predictions from classical molecular dynamics (MD) computer simulations rely on fitting empirical forcefields to match the experimental data, and very often assume the rigid water molecule and neglect the fluctuation of electric polarizability with the local chemical environment. The studies by classical MD simulations are scattered in the predicted physical trend, therefore inconclusive[26,31,32]. However, ab initio molecular dynamics[33] (AIMD) provides an ideal framework to predict the structures of electrolytes from quantum mechanical principles. In AIMD, the electronic structure of the ground state is generated on the fly from density functional theory[34] (DFT). Due to the delicate nature of the hydrogen-bond in water, much more expensive AIMD simulations, carried out at a higher level of electron exchange-correlation approximation, are demanded here than for ordinary materials[35–37]. Therefore, AIMD simulations on salt solutions performed so far have been limited to one or two selected concentrations[20,28,38].

To overcome the aforementioned limitation of AIMD, in this paper, we perform MD simulations on NaCl, KCl, and sodium bromide (NaBr) solutions and pure water via the deep potential approach[39]. In the main text of this paper, we have used the theoretical results obtained from NaCl solution as a prototypical example of Hofmeister series ions to elucidate the nature of the pressure-like effect. The physical pictures from simulated KCl and NaBr solutions are rather similar to the NaCl solution, and therefore the same conclusion is drawn. For clarity of the presentation, we have presented all the results of KCl and NaBr solutions in Supplementary Discussion. The needed deep neural network potential is trained on high-level DFT calculations based on the strongly constrained and

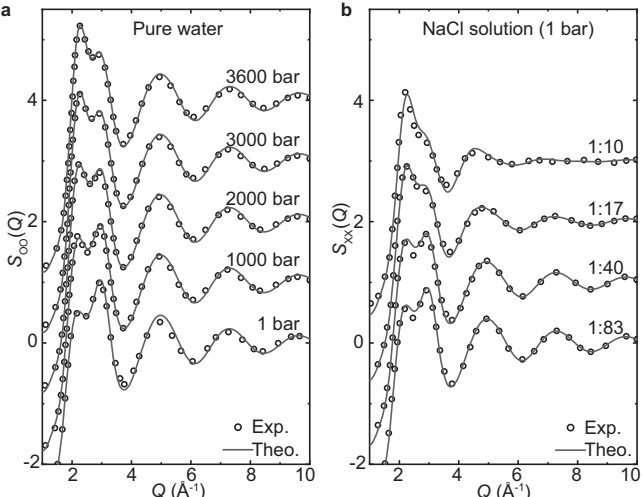

**Fig. 1 Reciprocal-space structure of pure water and NaCl solutions. a** Experimental[41,52] and theoretical partial structure factors $S_{OO}(Q)$, for O atoms in pure water at different pressures. **b** Experimental[18] and theoretical composite partial structure factors $S_{XX}(Q)$ for NaCl aqueous solutions with salt: water mole ratios 1:83 to 1:10, at 1 bar. $S_{XX}(Q)$ contains O–O, O–ion, and ion–ion correlations, $S_{XX}(Q) = \sum_{\alpha,\beta} w_{\alpha\beta} S_{\alpha\beta}(Q)$, $(\alpha, \beta \in \{O, Na, Cl\})$, where $\alpha$, $\beta$ are the atom types and $w_{\alpha\beta}$ is the corresponding weight. All structure factors were shifted vertically for visual clarity.

appropriately normed (SCAN) functional[40], which satisfies all 17 known exact constraints on semi-local exchange-correlation functionals and includes intermediate-ranged van der Waals interactions inherently without any empirical parameter. Deep potential MD allows us to study liquids at the accuracy of DFT but with significantly reduced computational cost. The methodology employed is described in Supplementary Methods.

## Results and discussion

**Liquid structure in reciprocal space**. The structure factor determined in diffraction experiments, by measuring the differential cross-section as a function of momentum transfer, $Q$, contains details on the correlation of the ions and water molecules in reciprocal space. Equivalently, the structure factor can also be obtained by inverting the real-space correlation functions from an equilibrated MD electrolyte structure into reciprocal space. In Fig. 1a, b, we present the experimental and theoretical structure factors $S_{OO}(Q)$ of neat water, as a function of pressure, along with the $S_{XX}(Q)$ of NaCl solutions at various concentrations, respectively. Importantly, the excellent agreement between experiment and theory gives support to computational methodologies being employed. In neat water, strong modulations on the structure factor by applied pressure have been repeatedly reported[41,42]. As shown in Fig. 1a, the first peak of $S_{OO}(Q)$ becomes higher with elevated pressure, while an attenuation is observed on its second peak. Notably, the structure factor of NaCl solutions in Fig. 1b exhibits rather similar behavior, however as a function of increased salt concentration. For pure water, the prominent changes of $S_{OO}(Q)$ under high pressures are caused by drastic distortions of the tetrahedral structure throughout the liquid[41,42]. Because of the similar pressure-like effect, it has been widely assumed that the water structure in NaCl solutions is distorted in the same way since $S_{XX}(Q)$ is mostly determined by the partial structure factors $S_{OO}(Q)$[17,18].

**Liquid structure in real space**. To unveil a physical picture, the observed pressure-like effect in the structure factors needs to be

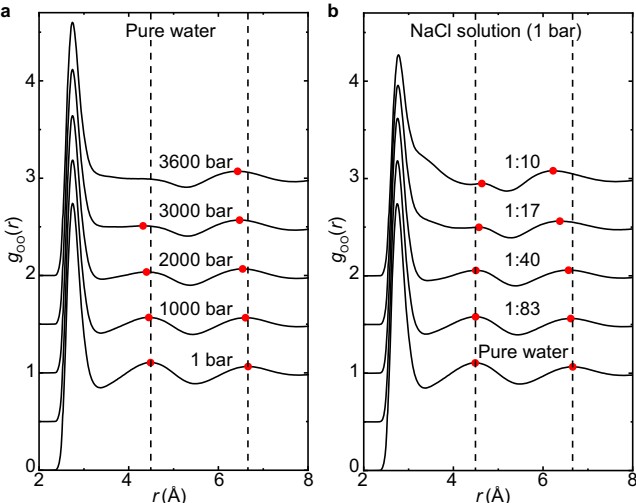

**Fig. 2 Real-space structures of pure water and NaCl solutions. a** Calculated O–O radial-distribution function, $g_{oo}(r)$, of pure water at different pressures. **b** Calculated $g_{oo}(r)$ of NaCl solutions for different concentrations at 1 bar. All RDFs were shifted vertically for visual clarity. The red dots indicate second and third peak positions; vertical dashed lines are for pure water at 1 bar.

examined in real space. The water-water correlation in real space is described by the oxygen–oxygen (O–O) radial-distribution function (RDF), $g_{oo}(r)$, in Fig. 2, which can be converted back into $S_{OO}(Q)$ through Fourier transformation. Under pressure, water becomes a denser liquid. The more compact structure is achieved by the inwards movements of both the second and third coordination shells in Fig. 2a, with a simultaneously increased population of interstitial water between the first and second coordination shells. The first coordination shell, as determined by the directional hydrogen bonding, barely changes. In addition, the intensity of the second coordination shell suffers a significant reduction, and even disappears at the highest pressure. The second coordination shell captures the tetrahedral molecule geometry with a characteristic distance of ~4.5 Å. Its collapse coincides with the well-established concept that the water structure is a highly distorted tetrahedral network under high pressure. The $g_{oo}(r)$ of NaCl solutions in Fig. 2b shares some common features with water under elevated pressure. As salt concentration increases, the third coordination shell also moves inwards together with a diminishing feature of the second coordination shell and more populated interstitial region. These similarities are reminiscent of the aforementioned pressure-like effect in reciprocal space. Nevertheless, there is a major difference in NaCl solutions; the second shell moves outwards, in the opposite direction to neat water under increased pressure. Because the water arrangement in this region is a critical signature of the tetrahedral ordering in water, the above difference suggests that the pressure-like effects in NaCl solutions likely have a distinct microscopic origin from water under high pressures.

**Microscopic structural insights**. At the molecular level, the structures of NaCl solutions are determined by the arrangement of the water molecules populating in ions' various solvation shells. In the FSS, water is bonded to both the ions and waters in the second shell. The nature of the interaction is determined by the electronic properties of both water and ions. On the one hand, the electronic orbitals in a water molecule adopt the well-known $sp^3$ hybridization giving rise to the near-tetrahedral structure of water, which manifests as the 4.5 Å O–O characteristic distance in

the RDF and the 109.5° O–O–O triplet angle in the angular distribution function (ADF), $P_{OOO}(\theta)$. On the other hand, $Na^+$ and $Cl^-$ both have a closed-shell electronic configuration. Therefore, neither of these ions has preferred bonding directions like a water molecule. Moreover, at room temperature, the water molecules surrounding the ions usually fluctuate among several competing solvation complexes.

The sodium cation attracts the negatively charged oxygen lone pair electrons of water via electrostatic interactions. Because of its small ionic radius of 1.02 Å, relatively tight solvation complexes are formed, composed mainly of triangular bipyramid, square pyramid, and square bipyramid[43] as schematically shown in Fig. 3a. The O–O distance probability distribution of the FSS water, $P_{OO}^{FSS}(r)$, has a rather structured distribution as shown in Fig. 3c. The major peak at 3.3 Å and the subpeak around 4.7 Å are contributed by the typical distances between the two nearest neighboring and the next-nearest-neighboring vertices on the polyhedra, respectively. Similarly, the ADF of water molecules in the FSS of $Na^+$ also loses the tetrahedral characteristic at 109.5°. As shown in Fig. 3f, the $P_{OOO}^{FSS}(\theta)$ of $Na^+$ has a structured distribution with two peaks centered at 70° and 99°, determined by the geometries of the characteristic polyhedra.

The chloride anion is hydrogen-bonded to the positively charged water protons. However, the $Cl^-$ has a sizable FSS due to its relatively large ionic radius of 1.81 Å. The water molecules encompassing the anion adopt a rich variety of solvation complexes, which are comprised of polyhedra with five to ten vertices. Again, the solvation complexes are distinct from the tetrahedra found in neat water. Water molecules in the FSS of $Cl^-$ are separated by the characteristic distances applicable to the polyhedra of its solvation complexes, which is a broad distribution function centered at 4.3 Å and 6.1 Å in Fig. 3c. The $P_{OOO}^{FSS}(\theta)$ of $Cl^-$ in Fig. 3f also deviates from that in neat water, and it contains features centered ~80° and 115°. The broad distance and angular distributions of O atoms in the FSS of $Cl^-$ are also consistent with more flexible solvation structures than those of $Na^+$.

In NaCl solutions, both cations and anions are present. Their FSSs collectively displace the liquid structures away from that of bulk water. As shown in the $g_{oo}(r)$ in Fig. 3b, the seemingly increased population of interstitial water and the outwards movement of the second coordination shell are mainly attributed to the FSS of $Na^+$, whereas the solvation complex of $Cl^-$ should be responsible for the apparent inward movement of the third coordination shell of solutions. The above effects become more significant when NaCl concentration increases, as shown by the shaded areas in Fig. 3b, because the relative number of water molecules that belong to ionic FSSs increases with concentration (the fraction of water molecules in FSSs are listed in Supplementary Table 1). A similar scenario also applies to the $P_{OOO}(\theta)$. With increased salt concentration, $P_{OOO}(\theta)$ shifts away from a tetrahedral distribution towards smaller angles. In pure water, a similar effect on the $P_{OOO}(\theta)$ has been observed at elevated pressures[44]. However, they have distinct microscopic origins. In NaCl solutions, it is caused mainly by the characteristic $P_{OOO}^{FSS}(\theta)$ distribution of ions, in particular by those of $Na^+$ as shown in Fig. 3f. In sharp contrast, the effect observed in water under high pressure is due to highly distorted tetrahedra throughout the entire liquid[41,42].

The water molecules located beyond ionic FSSs can be considered as free waters (FW) since they are not directly bonded by the ions. In this region, the molecules in the bulk solvent start to build back the hydrogen-bond network. To examine the extent of the structural restoration, a comparison of water-water distance distributions can be made between the

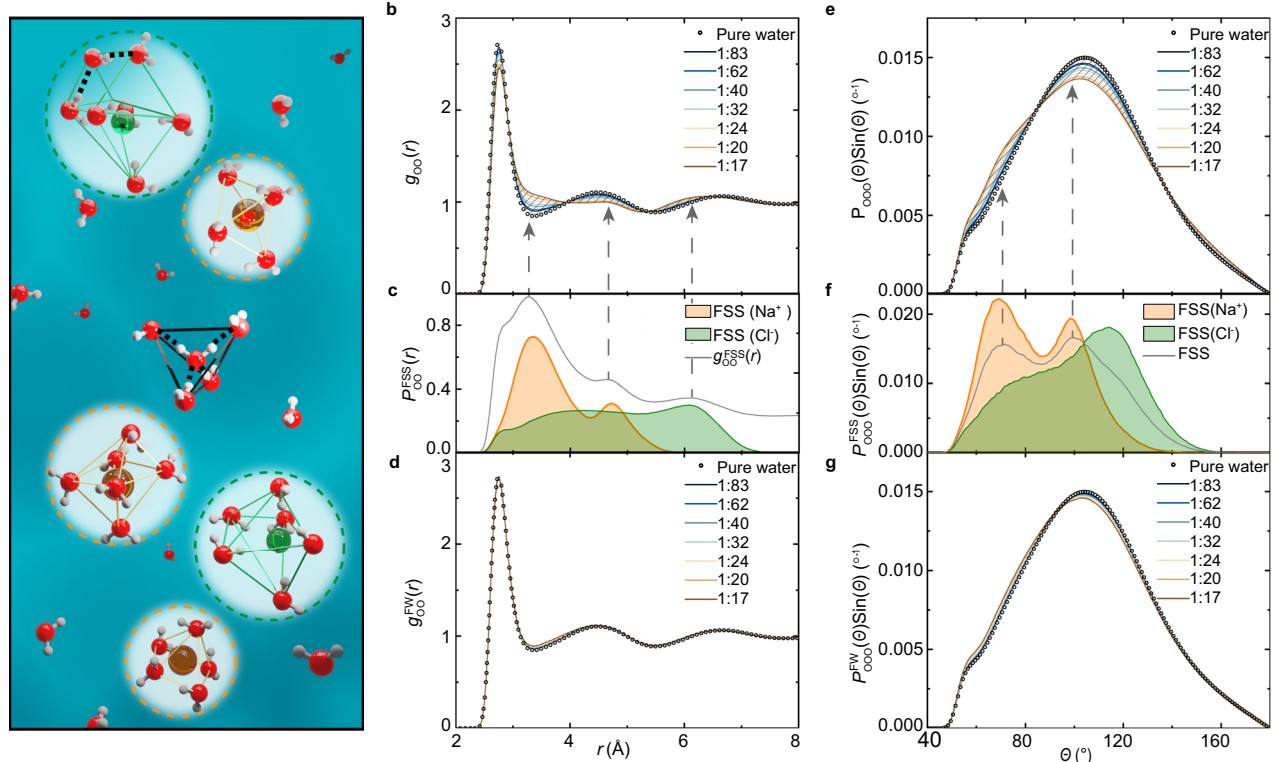

**Fig. 3 Structure of NaCl solutions. a** Schematic diagram of the microscopic structure of the NaCl solution. Red, white, yellow, and green spheres represent the O, H, $Na^+$, and $Cl^-$ atom/ion, respectively. The dashed yellow/green circles show FSSs of $Na^+$/$Cl^-$, the radius of which was defined as the first minimum of the O–Na/O–Cl RDF. The dashed black lines represent hydrogen bonds between water molecules. The hydrogen-bond chain in the FSS of $Cl^-$ indicates the partial recovery of the tetrahedral structure of water. **b** O–O RDFs, $g_{oo}(r)$, for pure water and NaCl solutions at various indicated concentrations. **c** O–O distance probability distributions in the FSS of a $Na^+$ or $Cl^-$ ion, $P_{oo}^{FSS}(r)$. The gray line $g_{oo}^{FSS}(r)$ is the contribution of all FSSs to $g_{oo}(r)$. The absolute value of $g_{oo}^{FSS}(r)$ is scaled. **d** O–O RDFs of FW, $g_{oo}^{FW}(r)$, for NaCl solutions compared with $g_{oo}(r)$ for pure water. **e** O–O–O ADFs, $P_{OOO}(\theta)$, for pure water and NaCl solutions. **f** O–O–O ADFs, $P_{OOO}^{FSS}(\theta)$, for the FSSs of $Na^+$, $Cl^-$, and both $Na^+$ and $Cl^-$. **g** O–O–O ADFs of FW, $P_{OOO}^{FW}(\theta)$, for NaCl solutions compared with $P_{OOO}(\theta)$ for pure water. The 1:10 results are not presented because the excluded volume correction is inapplicable[17]. Magnified differences of NaCl solutions with respect to neat water are presented in Supplementary Fig. 15.

solutions and neat water. To this end, the O–O RDF should be calculated in the absence of the ionic FSSs, i.e., only the FW are accounted, which is referred to as $g_{oo}^{FW}(r)$. In such an analysis, cavities are generated in the solution by excluding the FSSs. In order to compare the RDFs obtained from continuous pure water with that from the solutions with void regions, excluded volume corrections are implemented in computing the $g_{oo}^{FW}(r)$ as introduced by Soper et al.[17,45]. The resulting $g_{oo}^{FW}(r)$ of NaCl solutions are presented in Fig. 3d together with the $g_{oo}(r)$ of neat water. For all the concentrations under consideration, $g_{oo}^{FW}(r)$ largely recovers the bulk structure of neat water. The reinstatement of the tetrahedral ordering is further corroborated by the $P_{OOO}^{FW}(\theta)$ computed in the same region, which has the characteristic tetrahedral angle of around 109.5°.

Although visible differences can still be seen between the FW structures in the NaCl solutions and neat water in Fig. 3g, the overall tetrahedral network does not undergo drastic distortions outside the FSS in the solutions. Notably, the collective inwards movement of both the second and third coordination shells, which are the structural signatures of neat water under high pressures, are not present in the FWs of NaCl solutions. Rather similar pictures are also seen in the simulated KCl and NaBr solutions, the details of which are presented in Supplementary Discussion.

We want to stress that the current work focuses on elucidating that the pressure-like effect observed in salt solutions is not equivalent to the drastically distorted H-bond network by

applying thousands of atmospheres pressure on the neat water. The pressure-like effect was originally observed in structural factors of solutions probed by neutron-scattering experiments, and the structural factor is directly associated with the radial-distribution functions in real space via a Fourier transform. In recent years, whether or not the H-bonded network is fundamentally changed by dissolved salt has been at the center of scientific interest. To address the above question, more delicate order parameters beyond the radial-distribution function, such as the orientational correlation, should be carefully analyzed. In this work, we also investigated the orientational correlation function in dilute NaCl solutions in which the long-range Coulombic interaction is included as well. We found that the delicate orientation correlation in dilute solutions deviates from that of bulk water, which is in qualitative agreement with the results reported in literature[25,46–48]. We provide the detailed results and analyses in Supplementary Discussion.

**Virtual diffraction experiments**. The molecular structure of NaCl solutions in real space also provides important clues to elucidate the nature of the pressure-like effect in structure factors measured in reciprocal space. In this regard, the structure factors are required to be separated by the range and be unambiguously assigned into contributions from different solvation shells. Unfortunately, such a goal cannot be achieved by current experimental techniques. However, this difficulty can be resolved by performing virtual diffraction experiments based on our

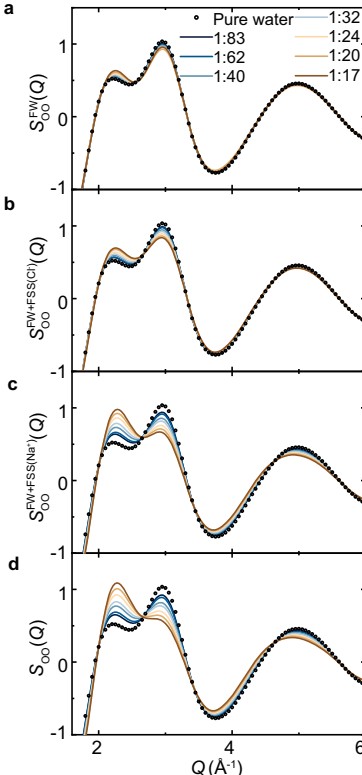

**Fig. 4 Virtual diffraction experiments of NaCl solutions.** The O–O structure factors contributed by **a**, the FW ($S_{OO}^{FW}(Q)$). **b** The FW and the FSS of Cl$^-$ ($S_{OO}^{FW+FSS(Cl-)}(Q)$). **c** The FW and the FSS of Na$^+$ ($S_{OO}^{FW+FSS(Na+)}(Q)$). **d** All water molecules in the system ($S_{OO}(Q)$). The 1:10 results are not presented because the excluded volume correction is inapplicable[17].

computer simulations. We start with a virtual diffraction experiment, in which it only measures the structure factors of solutions in the absence of the ionic FSSs, i.e., $S_{OO}^{FW}(Q)$. In computer simulations, $S_{OO}^{FW}(Q)$ can be straightforwardly calculated through a Fourier transformation of $g_{oo}^{FW}(r)$ in Fig. 3d. For all the NaCl concentrations under consideration, a nearly quantitatively agreement can be identified between the $S_{OO}^{FW}(Q)$ of NaCl solutions and the $S_{OO}(Q)$ of pure water as shown in Fig. 4a. The complete disappearance of the pressure-like effect in $S_{OO}^{FW}(Q)$ is consistent with the aforementioned fast recovery of the hydrogen-bond network of water outside the ionic FSSs. It further provides strong evidence that the pressure-like effects observed in NaCl solutions and in water under high pressures do not share the same structural origin at the molecular level.

Clearly, ionic FSSs are responsible for the pressure-like effect in NaCl solutions. In order to clarify the roles played by individual types of ions, we perform virtual diffraction experiments facilitated by the excluded volume correction methods[17,45]. In these computer experiments, the structure factors of $S_{OO}^{FW+FSS(Na+)}(Q)$, $S_{OO}^{FW+FSS(Cl-)}(Q)$, $S_{OO}(Q)$ are simulated by adding the FSSs of sodium cations, chloride anions, and both cations and anions systematically back onto the FW. As expected, the pressure-like effect reappears after considering the FSSs of both ions in the computed structure factor, Fig. 4d. However, compared to the weak modulation on the first two leading peaks in the structure factor by hydrated Cl$^-$ in Fig. 4b, the hydrated cation has a much more detrimental effect, as shown in Fig. 4c. The minor role played by Cl$^-$ is consistent with its flexible FSS structure, in which 28% of water molecules are not hydrogen-bonded by the Cl$^-$ anion. These nonbonded

FSS water molecules are likely to be attracted to each other via hydrogen bonding, and can further form hydrogen-bond chains with a third water molecule nearby, as schematically shown in Fig. 3a. Indeed, nearly 15% of water molecules in anion's FSS form hydrogen-bond chains. The formation of hydrogen-bond chains reflects the fact that hydrated Cl$^-$ is less destructive to the underlying water structure, which starts to partially recover even in the anion's FSS. In contrast, a similar scenario rarely occurs in the FSS of Na$^+$ because of its rigid solvation structure, in which only less than 3% of water molecules can form hydrogen-bond chains. Not surprisingly, the FSS of the Na$^+$ is much more dissimilar to the structure of liquid water.

In conclusion, neural network potentials fitted to state-of-the-art DFT data, reproduce the concentration dependence of neutron-scattering structure factors from NaCl, KCl, and NaBr solutions. Our simulations provide evidence in real space that the pressure-like effects in the structure factors originate from changes in the solvent-water's topology caused by the intrusion of ionic FSSs. Notably, the water structure is not drastically revised in the same way as pure water undergoes by applying external pressure as large as thousands of atmospheres.

## Data availability

The complete DFT training data and the deep potential models for NaCl, KCl, and, NaBr solutions generated in this study have been deposited in the Figshare database[49].

## Code availability

Deep potential molecular dynamics simulations were conducted using the DeePMD-kit package[39] (https://github.com/deepmodeling/deepmd-kit) in conjunction with LAMMPS[50] (https://www.lammps.org/). The deep potential model was trained using an active machine learning approach called deep potential generator[51] (https://github.com/deepmodeling/dpgen). The code used to generate the plots shown in the main text is available from the corresponding author upon request.

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

## Acknowledgements

We thank Roberto Car, Linfeng Zhang, and Han Wang for fruitful discussions. This work was supported by the "Chemistry in Solution and at Interfaces" (CSI) Center funded by the U.S. Department of Energy through Award No. DE-SC0019394 (C.Z., S.Y., A.Z.P., M.L.K., and X.W.). We also acknowledge support from the National Science Foundation through Award No. DMR-2053195 (X.W.). This research used resources of the National Energy Research Scientific Computing Center (NERSC), which is supported by the U.S. Department of Energy (DOE), Office of Science under Contract No. DE-AC02-05CH11231 (C.Z. and X.W.). This research includes calculations carried out on HPC resources supported in part by the National Science Foundation through major research instrumentation grant number 1625061 and by the U.S. Army Research Laboratory under contract No. W911NF-16-2-0189 (M.L.K.). This research used resources of the Oak Ridge Leadership Computing Facility at the Oak Ridge National Laboratory, which is supported by the Office of Science of the U.S. Department of Energy under Contract No. DE-AC05-00OR22725 (C.Z., S.Y., A.Z.P., and X.W.).

## Author contributions

X.W. and M.L.K. designed the project. C.Z. and S.Y. carried out the simulations and performed the analysis. X.W., M.L.K., A.Z.P., C.Z., and S.Y. wrote the manuscript.

## Competing interests

The authors declare no competing interests.
