## [Peer Review File · Nature Communications]

REVIEWERS' COMMENTS

Reviewer #1 (Remarks to the Author):

The authors have addressed all the points raised in my initial review. The new version clarifies some claims and includes a substantial amount of new results (mainly in the Supplementary Information) with additional salts and analyses (in particular on the orientational order). I also note that changes made in response to my comments also address those of other reviewers, in particular from Reviewer #2 (even though expressed differently, there was a large overlap with mine, see their comments 1, 4 and 5). Important additions in the revised version are the results for other salts (provided as Supplementary Information, since the conclusions are similar), which was the topic of the 2nd comment of Reviewer #2, and the discussion of nuclear quantum effects, in response to their 3rd comment.

I am therefore happy to recommend publication of this work in Nature Communications in the present form.

Comments from Referee #1 and our detailed response

Comment: “The authors have addressed all the points raised in my initial review.

The new version clarifies some claims and includes a substantial amount of new results (mainly in the Supplementary Information) with additional salts and analyses (in particular on the orientational order). I also note that changes made in response to my comments also address those of other reviewers, in particular from Reviewer #2 (even though expressed differently, there was a large overlap with mine, see their comments 1, 4 and 5). Important additions in the revised version are the results for other salts (provided as Supplementary Information, since the conclusions are similar), which was the topic of the 2nd comment of Reviewer #2, and the discussion of nuclear quantum effects, in response to their 3rd comment.

I am therefore happy to recommend publication of this work in Nature Communications in the present form.”

Author reply: We thank referee #1’s recommendation. We also appreciate referee #1’s initial review, which is important for improving the manuscript.